# Phagocytosis of Erythrocytes from Gaucher Patients Induces Phenotypic Modifications in Macrophages, Driving Them toward Gaucher Cells

**DOI:** 10.3390/ijms23147640

**Published:** 2022-07-11

**Authors:** Lucie Dupuis, Margaux Chauvet, Emmanuelle Bourdelier, Michaël Dussiot, Nadia Belmatoug, Caroline Le Van Kim, Arnaud Chêne, Mélanie Franco

**Affiliations:** 1INSERM, UMR_S1134, BIGR, Université Paris Cité and Université des Antilles, F-75015 Paris, France; lucie.dupuis@pasteur.fr (L.D.); margauxchauvet@hotmail.fr (M.C.); emmanuelle.bourdelier@inserm.fr (E.B.); caroline.le-van-kim@inserm.fr (C.L.V.K.); arnaud.chene@inserm.fr (A.C.); 2UMR_S1163, Sorbonne Paris Cité, Institut Imagine, Laboratoire d’Excellence GR-Ex, Université Paris Cité, F-75015 Paris, France; michael.dussiot@gmail.com; 3APHP CRML Maladies Lysosomales, Service de Médecine Interne, Hôpital Beaujon, Sorbonne Université, F-92110 Clichy, France; nadia.belmatoug@aphp.fr

**Keywords:** red blood cells, erythrophagocytosis, macrophages, Gaucher disease

## Abstract

Gaucher disease (GD) is caused by glucocerebrosidase deficiency leading to the accumulation of sphingolipids in macrophages named “Gaucher’s Cells”. These cells are characterized by deregulated expression of cell surface markers, abnormal secretion of inflammatory cytokines, and iron sequestration. These cells are known to infiltrate tissues resulting in hematological manifestations, splenomegaly, and bone diseases. We have already demonstrated that Gaucher red blood cells exhibit altered properties suggesting their key role in GD clinical manifestations. We hypothesized that Gaucher’s erythrocytes could be prone to premature destruction by macrophages contributing to the formation of altered macrophages and Gaucher-like cells. We conducted in vitro experiments of erythrophagocytosis using erythrocytes from Gaucher’s patients or healthy donors. Our results showed an enhanced erythrophagocytosis of Gaucher red blood cells compared to healthy red blood cells, which is related to erythrocyte sphingolipids overload and reduced deformability. Importantly, we showed elevated expression of the antigen-presenting molecules CD1d and MHC-II and of the iron-regulator hepcidin in macrophages, as well as enhanced secretion of the pro-inflammatory cytokine IL-1β after phagocytosis of GD erythrocytes. These results strongly suggested that erythrophagocytosis in GD contribute to phenotypic modifications in macrophages. This present study shows that erythrocytes-macrophages interactions may be crucial in GD pathophysiology and pathogenesis.

## 1. Introduction

Gaucher disease (GD) is the most common human genetic lysosomal storage disorder, which is caused by a recessive mutation in the gene coding for the β-glucocerebrosidase (GCase). GD is classified into three types. Type 1, the most common pathological form of GD, is characterized by hepato-splenomegaly, anemia, thrombocytopenia, complex bone disorders (osteonecrosis with bone micro-infarcts in ischemic regions, and avascular necrosis), [1,2,3] and an absence of neurological disorders. Type 1 GD is efficiently treated with enzyme replacement therapy (ERT), which consists of the infusion of recombinant GCase, which significantly reduces clinical manifestations [4]. Unlike type 1, types 2 and 3 GD have early-onset brain involvement getting worse over time for the affected individuals. There is to date no treatment that can efficiently act on the evolution of the neurological consequences.

GCase is essential for the degradation of sphingolipids within lysosomes and its deficiency causes sphingolipids accumulation, especially in macrophages. Lipid-laden macrophages display alternatively activated phenotype and some transform into Gaucher’s cells (GC). These GC present a specific phenotype, mainly studied by immunohistochemistry of bone marrow samples, and display the striated appearance of a “wrinkled tissue paper” due to lipid and iron overload [5]. Alternatively, activated-macrophages and GC infiltrate tissues within different organs such as the liver, the spleen, and the bone marrow. Together, the alterations of the monocytic lineage and the rise of GC have been considered responsible for the major signs of the disease. However, the hematological manifestations as well as hepatosplenomegaly and bone disorders strongly suggested that red blood cells (RBCs) could also be important actors in pathophysiology. Indeed, various alterations of RBCs from patients with GD have been previously reported, such as abnormal morphologies, increased adhesion properties, reduced deformability [6], and sphingolipids accumulation within the cells, i.e., glucosylceramide (GL-1), glucosylsphingosine (Lyso-GL-1), sphingosine (Sph), sphingosine-1-phosphate (S1P) [7,8,9]. Notably, many of these alterations were reduced or normalized after treatment of the patients with ERT [9]. One of our recent studies suggests that sphingolipids accumulation occurs both in erythroid progenitors during erythropoiesis as well as in mature RBCs in the peripheral circulation [8].

RBCs are able to physically interact with macrophages at different developmental stages throughout their life; at their genesis, during late erythropoiesis within the erythroblastic islands, and at senescence, when RBCs undergo the clearance process, being captured and phagocytized by resident macrophages within the red pulp of the spleen.

Macrophages are professional phagocytes and antigen-presenting cells. However, they exhibit various phenotypes differing in their site of location and depending on the cytokines secreted by the surrounding cells [10]. Macrophages are routinely classified into two distinct subpopulations: the classically activated type 1 (M1), and the alternatively activated type 2 (M2). M1-macrophages are characterized by an inflammatory phenotype (secretion of pro-inflammatory cytokines) and the M2 subtype, which participates in the resolution of inflammation and in tissue homeostasis, is characterized by the production of anti-inflammatory cytokines and expresses receptors of innate immunity. In vitro, macrophage-like cells (uncommitted macrophages, without specific polarization) can be generated from the human monocytic leukemia cell line THP1 (THP1-derived macrophages) [11]. In GD, GC and the macrophages are alternatively activated and the expression of several cell surface markers, as well as the secretion of pro- and anti-inflammatory cytokines/chemokines, are dysregulated [12,13,14] as compared to normal macrophages.

Two studies reported that the half-life of RBCs was decreased in GD type 1 patients [15,16]. Additionally, histological experiments performed on bone marrow smears from Gaucher’s patients, identified abnormal erythrophagocytosis in this compartment, where RBCs and macrophages highly interact [17,18,19,20]. However, these observations have been reported on a limited number of patients and our knowledge of the resulting biological consequences from these observations thus remains sparse.

We hypothesized here that the abnormal properties of Gaucher’s RBCs could induce their premature uptake by macrophages, leading to the formation of altered macrophages and GC, and thus, to the spreading of the disease. By developing an in vitro erythrophagocytosis assay using THP1-derived macrophages or primary macrophages (derived from circulating monocytes) polarized into M2 phenotype, we evidenced an enhanced phagocytosis of type 1 GD RBCs by normal macrophages, which is governed by alterations in GD RBC membrane properties.

We also showed that erythrophagocytosis of lipid-laden GD RBCs induce phenotypic alterations in normal macrophages, up-regulating the expression of inflammatory molecules, such as the scavenger receptor CD36, the antigen-presenting molecules CD1d, and MHC-II, the hepcidin (systemic regulator of iron stock) and the secretion of the pro-inflammatory cytokine IL-1β. These markers are known to be up-regulated in GC as well.

Taken together, our results support a model in which phagocytosis of GD RBCs by macrophages induces phenotypical changes in these cells, driving them toward GC. RBCs could therefore act as crucial players in GD pathophysiology and pathogenesis.

## 2. Results

### 2.1. Preferential Phagocytosis of GD RBCs by Macrophages

To reliably monitor phagocytosis of RBCs in vitro, an optimized assay was developed, in which carboxyfluorescein succinimidyl ester (CFSE)-labeled RBCs (Appendix A) were co-incubated with THP1-derived macrophages [11] or with primary macrophages derived from monocytes freshly isolated from a blood donor and polarized into M2 phenotype. In this type of assay, a fluorescence transfer from RBCs to phagocytic cells reflected erythrophagocytosis.

The potency of macrophages to efficiently performtheir phagocytic functions was controlled in each set of experiments by introducing a condition in which RBCs were heated at 50 °C. Indeed, the heating of RBCs at high temperatures considerably affects the biomechanical properties of the cells and leads to substantial erythrophagocytosis by competent cells as reflected by an elevated percentage of CFSE^+^ macrophages (51.4%) as compared to the condition in which RBCs were not heated (12.5%) (Appendix A). To ensure that the CFSE signal gained by the macrophages only resulted from erythrophagocytosis and not from RBCs-macrophages conjugates, the samples were subjected to an osmotic shock at the end of the incubation period to remove all non-phagocytized erythrocytes. The data obtained from imaging flow cytometry analysis presented in Appendix A confirmed the absence of RBCs-macrophages conjugates after osmotic shock. Notably, the values of geometrical mean fluorescence intensity (GeoMFI) of the CFSE^+^ macrophage population positively correlated with the number of fluorescent (CFSE) spots present within the phagocytic cells (reflecting the number of engulfed RBCs) (Appendix A). In order to ensure reliability, the data were normalized and expressed as a phagocytosis index (PI) (see Methods S1). Next, in vitro phagocytosis was performed using macrophages derived from fresh monocytes or from THP1 and RBCs from healthy donors or GD patients treated or not by ERT.

We first carried out an autologous erythrophagocytosis assay using blood-derived macrophages from three healthy donors (control group) or three GD patients, and RBCs originating from the corresponding individuals. Although the number of samples in the experiment is too low to perform a robust statistical analysis, the PI appeared higher for the GD group as compared to the control group (median PI = 104 and PI = 206.3, respectively), thus reflecting a larger number of GD RBCs internalized by macrophages (Figure 1A).

We then performed an erythrophagocytosis assay using blood-derived macrophages isolated from healthy donors (polarized toward M2 phenotype) and RBCs from either GD patients or control (CTL) RBCs. Phagocytes were treated or not with the GCase inhibitor (Conduritol B Epoxyde; CBE) six days prior to the experiment to mimic the GCase deficiency present in GD. We observed a higher PI when GD RBCs were co-incubated with macrophages either untreated or CBE-treated macrophages (median PI = 148.3 and PI = 182.8, respectively) as compared to RBCs from healthy donors (median PI = 96.81 and PI = 101.5, respectively), reflecting a preferential uptake of GD RBCs by phagocytic cells (Appendix A). In addition, the PI of GD RBCs appeared similar, regardless of the type of phagocytic cells used in the assay (control or Gaucher-like macrophages).

In order to investigate the effect of ERT treatment on GD RBCs, we then carried out an erythrophagocytosis assay using monocyte-derived macrophages polarized toward M2 phenotype originating from healthy donors and RBCs from GD patients treated or not by ERT (Figure 1B). The use of GD RBCs from untreated GD patients led to a significantly higher PI (median PI = 134.2) as compared to the use of CTL RBCs (median PI = 110.8) (*p* = 0.0464). On the other hand, the PI of the ERT GD RBCs group (median PI = 92.38) was closed to one of the CTL RBCs (median PI = 110.8) (*p* = 0.0649).

In order to confirm our results with another model, we used the human monocytic cell line THP1, which is a well-established model and a valuable tool to study macrophage function in controlled conditions, notably by limiting inter-individual variations inherent to the use of primary cells [21]. THP1-derived macrophages were co-incubated with CTL and GD RBCs. GD RBCs were issued from a well-characterized GD patient cohort (detailed demographic, clinical, and biological data are provided in Appendix A). The resulting PI was significantly higher for the GD RBCs group (median PI = 121) as compared to the CTL RBCs group (median PI= 83.22) (*p* = 0.0001) (Figure 1C left panel).

Erythrophagocytosis assays were then performed using THP1 cell line and GD RBCs from patients enrolled in a longitudinal study; before ERT (Pre-ERT RBCs) and 6 to 42 months after the beginning of ERT (Post-ERT RBCs) (see Appendix A for biological and clinical data). The results revealed that co-incubation of macrophages with Post-ERT RBCs led to a significantly lower PI (median PI = 107.1) than co-incubation with Pre-ERT RBCs (median PI = 121) (*p* = 0.0163) (Figure 1C right panel). These results are in line with those previously obtained with macrophages derived from primary cells (Figure 1A,B) and thus confirm the relevance of the THP1 model to further conduct our study.

Taken together, these results generated with primary cells and THP1 cell line, show an enhanced erythrophagocytosis of GD RBCs, most likely driven by intrinsic properties of GD erythrocytes.

### 2.2. The Uptake of the GD RBCs Is Independent of Phosphatidylserine Exposure, of CD47-Sirpα “Don’t Eat Me” Signal and of Opsonizing Immunoglobulins G

There is to date no real consensus on which RBC determinant(s) is(are) preponderant for mediating erythrocyte clearance by macrophages. Among the different erythrocyte biological changes potentially mediating their removal by phagocytic processes [22,23,24], cell surface exposure of phosphatidylserine (PS) and alteration of CD47-Sirpα interactions (“don’t eat me” signal) are particularly well investigated [24,25]. We thus assessed and compared the cell surface expression of PS and CD47 between RBCs originating from healthy donors and from GD patients. No statistically significant difference in PS nor CD47 surface expression between CTL and GD RBCs was detected by flow cytometry (Appendix A left panel, respectively). Since CD47 distribution within the plasma membrane also plays an important role in mediating inhibitory signaling to prevent phagocytosis, notably by forming clusters to ensure high avidity interactions with the regulatory membrane glycoprotein Sirpα [26], the binding of recombinant Sirpα to the surface of CTL and GD RBCs was also investigated. No significant difference in Sirpα binding was observed between the CTL and GD RBCs (Appendix A right panel).

The presence of immunoglobulins reacting to sphingolipids has been reported in GD [27]. Since RBCs opsonization by autoreactive IgG could potentially engage fraction crystallizable gamma receptor (FcγR) present at the surface of phagocytic cells (with the subsequent activation of cellular pathways leading to opsonic phagocytosis), we assessed the presence of surface-associated IgG on CTL and GD RBCs. Using flow cytometry, we could not evidence the presence of IgG bound to the surface of either type of RBCs (Appendix A).

### 2.3. The Uptake of GD RBCs Correlates with Their Altered Cellular Morphology and Impaired Deformability

In the absence of classical signals for RBCs’ internalization by phagocytes, the engulfment of GD RBCs most likely relies on another (others) mechanical process(es) directly linked to alterations in RBC membrane architecture remodeling, and/or in cellular deformability. To determine erythrocyte determinant(s)-mediating erythrophagocytosis, we cross-analyzed the data produced in Figure 1C (erythrophagocytosis assays with THP1 cells) with the biological parameters of the RBCs (reflecting RBCs’ morphologies and capacities of deformability) as previously described [8]. A positive correlation was observed between the PI and the percentage of abnormal RBCs’ morphologies (*p* = 0.0002) (Figure 2A left panel) accompanied by a negative correlation with RBCs’ deformability (*p* = 0.0031) (Figure 2A right panel).

We developed at the same time experimental models to confirm the impact of RBC rigidity on erythrophagocytosis. Mature RBCs from healthy donors were heated or fixed with glutaraldehyde to artificially reduce their deformability capabilities. As depicted in Figure 2B left panel, both treatments led to an increased phagocytosis of RBC as compared to non-treated RBCs and RBC fixation was more potent at inducing erythrophagocytosis than heating (median PI = 196.8 and PI= 154.3, respectively), (*p* = 0,0286 for all comparisons).

Furthermore, we demonstrate a negative correlation between the deformability of RBCs (reflected by the elongation index at 3Pa) and their uptake by the macrophages (PI) (*p* < 0.0001) (Figure 2B right panel). Thus, the less deformable the RBCs are, the more they are prone to be internalized by phagocytes.

Altogether, these results demonstrate a strong relationship between the morphological and biomechanical properties of RBCs and their susceptibility to being phagocytized by macrophages.

### 2.4. The Uptake of GD RBCs Correlates with Sphingolipids Overload

Our previous studies demonstrated that GD RBCs were overloaded in four sphingolipids; GL-1, Lyso-GL-1, Sph, and S1P [7]. This sphingolipids overload correlated with clinical parameters of the disease and with abnormal properties of GD RBCs, including a reduced cell deformability and a high level of atypical morphologies [7,9]. It was thus tempting to hypothesize that sphingolipids overload could, at least partly, contribute to enhanced erythrophagocytosis.

Correlation analysis was then performed between the PI obtained from the in vitro erythrophagocytosis assays (Figure 1C) and the sphingolipids levels of the RBCs (data originating from [7]). Positive correlations between the levels of sphingolipids measured in RBCs and the corresponding erythrophagocytosis indexes were highly significant for Lyso-GL-1, Sph, and S1P (*p* < 0.0001; *p* < 0.0001; *p* < 0.0001 respectively). Though statistically significant, the correlation between the GL-1 content of RBCs and the PI was less remarkable (*p* = 0.0124) (Figure 3A).

Additionally, we experimentally assessed the impact of lipid overload in erythroid cells on erythrophagocytosis. We here made use of a pre-established model, in which in vitro erythropoiesis is performed in the presence of CBE to mimic the production of reticulocytes overloaded with sphingolipids [8] (Figure 3B). After 18 days of in vitro erythropoiesis (with or without CBE), reticulocytes were sorted based on their positivity for glycophorin A (cell surface staining) and their negativity for the fluorescent DNA intercalant stain, Hoescht (Figure 3C left panel).

Purified reticulocytes (displaying or not sphingolipids overload) were co-incubated with THP1-derived macrophages. We observed a significantly higher PI of Gaucher-like reticulocytes (median PI = 142.6) as compared to control reticulocytes (median PI = 91.62) (*p* = 0.0317) (Figure 3C right panel).

Altogether, these results point to sphingolipids overload of GD RBCs and the consequent reduced deformability of the cells are potential determinants for induction of erythrophagocytosis.

### 2.5. The Uptake of GD RBCs Induces Phenotypic Modifications in Macrophages

Modification of the expression of surface molecules and of cytokines/chemokines production profiles, variations in iron metabolism regulation as well as oxidative bursts could strongly affect macrophage polarity, function, or survival [28,29,30]. We thus assessed potential phenotypic modifications in THP1-derived macrophages following phagocytosis of normal RBCs and GD RBCs.

For that, we measured by flow cytometry the expression of a variety of macrophage cell-surface markers including Sirpα, the macrophage polarization marker CD163, the scavenger receptor CD36, and CD1d, which is involved in lipid presentation and the major histocompatibility complex molecule MHC-II. After erythrophagocytosis, the same levels of Sirpα and CD163 were detected at the surface of macrophages having engulfed CTL or GD RBCs (Appendix A). The same levels of CD36, CD1d, and MHC-II were also observed between macrophages that engulfed CTL or heated healthy heated RBCs (Figure 4A). In contrast, a significant higher expression of CD36, CD1d and MHC-II was measured at the surface of macrophages which phagocytized GD RBCs as compared to CTL RBCs (*p* = 0.0009, *p* = 0.0038, *p* = 0.0003, respectively) (Figure 4A).

In addition, the production of a panel of pro and anti-inflammatory cytokines and chemokines was investigated after erythrophagocytosis. Out of the 7 molecules assessed, only the pro-inflammatory IL-1β levels increased in the culture medium of macrophages having ingested GD RBCs compared to CTL RBCs (*p* = 0.0182) (Figure 4B left panel). Comparing both conditions, the quantification of IL-10, TNF-α, MCP-1, and RANTES levels did not reveal any significant differences (Appendix A).

The transcription profile of the iron metabolism regulator hepcidin was also assessed. Hepcidin mRNA quantification was performed in macrophages after erythrophagocytosis of different RBC types. Interestingly, we observed a significant increase of hepcidin transcripts in macrophages having engulfed GD RBCs as compared to those having phagocytized CTL RBCs or heated RBCs (Figure 4B right panel).

The production of reactive oxygen species (ROS) was analyzed in undifferentiated THP1 cells, 3 h after the phagocytosis process (t0) as well as 24 h after the beginning of the assay (t + 24 h). ROS production in CFSE^+^ cells increased with time in both conditions (GD and CTL RBCs) but similar levels of ROS production were observed for a given timepoint for CTL RBCs and for GD RBCs (Appendix A, left panel). Profiling of heme-oxygenase1 (HO-1) transcription revealed a higher level of HO-1 mRNA 12 h (t + 12 h) after erythrophagocytosis as compared to t0. However similar mRNA levels were observed for a given timepoint for CTL and GD RBCs (Appendix A right panel).

Overall, the erythrophagocytosis of GD RBCs induced an increased expression of the scavenger receptor CD36, MHC-II, and the CD1d molecules. Moreover, the expression of hepcidin and the secretion of the pro-inflammatory cytokine IL-1β were also enhanced in macrophages having phagocytized GD RBCs as compared to CTL RBCs. Altogether, these results suggest that the phagocytosis of GD RBCs induces particular phenotypic modifications in macrophages.

### 2.6. The Enhanced Phagocytosis of GD RBCs Correlates with Markers of Disease Activity

In order to assess if the enhanced phagocytosis of GD RBCs described in this study could be linked to well-defined markers of GD activity in a physiological context, correlation analyses between the PI (Figure 1C) and hematological parameters (Appendix A) were carried out. Increased phagocytosis of GD RBCs by macrophages correlated with anemia (reflected by a low percentage of hematocrit) (*p* = 0.0208) (Figure 5 left panel) and with elevated plasma levels of Lyso-GL1 and ferritin (Figure 5, *p* <0.0001, *p* = 0.0267, for middle and right panel respectively). Thus, it appears that the more the patients are anemic, the more the RBCs are phagocytosed. We also observed a significant correlation between the plasma levels of Lyso-GL1 and anemia, but no correlation with ferritinemia (Appendix A). The strong relationship between plasma Lyso-GL1 levels and anemia had previously been observed in other studies [7]. This suggests a close relationship between the plasma marker Lyso-GL1 and the severity of anemia.

## 3. Discussion

Gaucher disease (GD) symptoms have been extensively described [31] and the contribution of macrophages and Gaucher cells (GC) in the disease has been well demonstrated. However, the precise biological mechanisms involved in the etiology of the complex pathophysiology affecting multiple organs still remain unclear. Beyond the contribution of macrophages in GD, several lines of evidence suggest that defects within the erythrocytic compartment may also participate in physiological dysfunctions and exacerbate clinical signs such as the emergence of ischemic and vaso-occlusive events. Indeed, RBCs from GD patients have been demonstrated to exhibit altered biological phenotypes such as enhanced aggregation and adhesive properties, abnormal morphologies, and reduced deformability [6,9]. Our recent studies suggest sphingolipids accumulation in GD RBCs as a main cause of erythrocyte structural/functional alterations [8,9]. We reasoned here that GD RBCs could play a more important role in pathogenesis than previously thought by interfering with key physiological processes leading to hallmark features in GD. Based on previous results showing that abnormal erythrophagocytic events occur in the bone marrow of GD patients [17,19], we hypothesized that altered RBCs properties could trigger erythrophagocytosis by macrophages and may affect their phenotype.

Although our experiments were carried out with different in vitro models of macrophages (primary cells isolated from fresh blood and polarized toward M2 phenotype or on THP1-derived macrophages), all our erythrophagocytosis assays showed that RBCs from patients with GD were more prone to phagocytosis than RBCs isolated from healthy individuals. Using RBCs samples from patients involved in a longitudinal study, we also showed that the PI decreased after ERT. The enhanced phagocytosis of GD RBCs appeared to be independent of classic molecular signals present at the target cell surface such as opsonization, an increase in PS expression, or a decrease in CD47-Sirpα “don’t eat me” signal. In contrast, the enhanced PI of GD RBCs strongly correlated with altered erythrocytes properties such as sphingolipids overload, reduced cellular deformability, and abnormal morphologies. This suggests that specific and altered membrane signals in GD RBCs could be responsible for their increased phagocytosis. Interestingly, by artificially inducing reduced deformability of the erythrocytes, we observed increased erythrophagocytosis by macrophages. Moreover, Gaucher-like reticulocytes generated from in vitro erythropoiesis in the presence of GCase inhibitor were more prone to phagocytosis as compared to normal erythrophagocytosis derived-reticulocytes. We then hypothesized that GD erythrocytes/reticulocytes could be internalized by a mechanical process due to lightened tension forces of the less deformable RBCs. Indeed, other studies already demonstrated the importance of the rigidity of the cellular target in the internalization process [32]. The authors of this later study demonstrated that the rigidity of fixed RBCs triggered their internalization independently of the CD47-Sirpα “don’t eat me” signal. We have previously evidenced that sphingolipids overload in RBCs membranes could affect their membrane deformability [7,9]. In addition, it has been demonstrated that the treatment of healthy RBCs with another type of lipid, phosphatidylcholine (LPC), induced dose-dependent alterations corresponding to reduced deformability, morphological changes, loss of surface area, and surface/volume ratio, and led to their retention into isolated-perfused human spleen [33]. Thus, sphingolipids overload leading to reduced deformability appears to be one, but not necessarily exclusive, candidate mediating erythrocyte clearance by macrophages in GD.

Although the statistical power of our study was not always sufficient to reach significance, we were able to observe increased erythrophagocytosis in ex vivo experiments using primary effector cells isolated from Gaucher patients. Previous studies brought evidence of abnormal erythrophagocytosis events in the bone marrow of GD patients [17,19]. It is tempting to hypothesize that such events could also occur in the spleen, whose functional role is to eliminate abnormal and senescent RBCs, thus participating in the splenomegaly observed in GD [22,34].

Correlations established from our in vitro results and hallmarks of GD strongly suggest that erythrophagocytosis could occur in the context of GD and participate in the pathophysiology. First of all, PI strongly correlated with the plasma Lyso-GL1 concentration, a major marker of GD activity. In addition, our study revealed a negative correlation between the PI and the hematocrit. These results suggest that abnormal erythrophagocytosis could contribute, at least partly, to the anemia observed in the disease. The PI also correlated with the hyperferritinemia observed in patients, implying that abnormal erythrophagocytosis could contribute to enhanced iron recycling leading to iron sequestration. In this line, the up-regulated expression of hepcidin observed after erythrophagocytosis of lipid-laden GD RBCs may also explain the mechanism of hyperferritinemia found in many patients with Gaucher disease [35].

Previous studies reported a role of GD IPSCs-derived macrophages models in the clearance of opsonized-sheep RBCs [14,36]. Our erythrophagocytosis results performed using GD RBCs and fully potent macrophages complement these previous studies and demonstrate that GD RBCs’ properties lead to their enhanced erythrophagocytosis independently of the alteration of macrophages. However, in GD patients, a clearance defect associated with enhanced uptake of GD RBCs could constitute more important storage of undegraded RBCs membranes in GD macrophages compared to our THP1 macrophage model. Thus, the altered phenotype of macrophages could be more severe in vivo and contribute to the deregulation of other markers such as the secretion of CCL18 and chitotriosidase activity.

We also investigated if the phagocytosis of GD RBCs could induce some phenotypic modification in THP1-derived macrophages potentially driving them toward a GC phenotype. We showed an increased surface expression of the CD36 phagocytic marker, the CD1d, and MHC-II immunological markers as well as an increased secretion of the cytokine IL1-β and the elevation of the hepcidin mRNA levels after phagocytosis of GD RBCs. Importantly, all of these markers are over-expressed in the context of GD [13,36,37,38]. IL-1β is known to contribute to inflammation in GD [39] and lipid presentation by CD1d leads to NKT cell activation. Lipid-specific NKT cells have been shown to induce germinal center B cell activation, resulting in the production of anti-lipid antibodies. The presence of anti-lipid antibodies in GD [40,41] raises the possibility that the enhanced non-opsonic erythrophagocytosis of GD RBCs that we observed in our study, could be exacerbated in vivo by antibody-dependent mechanisms (opsonic phagocytosis) and promote the modification of the macrophage’s phenotype [42], contributing to the formation of the GC. Interestingly, we previously showed in a GC model in which J774 macrophages were treated with CBE, an induced local hepcidin expression, suggesting iron sequestration and cell inflammation [43]. Increased phagocytosis of Gaucher RBCs might be envisaged as one, but not necessarily exclusive, cellular mechanism responsible for the origin of the iron sequestration observed in the pathology [15,44]. This is supported by a study performed on GD IPSCs-derived macrophages models, which suggested that a delay in efferocytosis function (phagocytosis and clearance of apoptotic cells) could contribute to splenomegaly and inflammation [45].

In summary, our results highlight that RBCs from patients with GD are prone to phagocytosis by macrophages and that the consequently altered macrophage phenotype could contribute to the genesis of GC. Further studies are needed to clarify the precise links between abnormal erythrophagocytosis of GD RBCs, GD clinical manifestations, and disease spreading.

Many lysosomal-storage diseases (Niemann-Pick disease type C, Fabry disease, and GD), as well as atherosclerosis, are characterized by abnormal lipid accumulation in macrophages suggesting a storage-compromised lysosomal system [46,47,48,49]. The observations, combined with the results of our work, call for additional studies assessing the impact of erythrophagocytosis on macrophage phenotypes, in a broader spectrum of pathologies for which lipid-laden cells are a hallmark. Targeting macrophages with an inhibitor of phagocytosis could thus prove to be an efficient approach to treating these types of pathologies.

## 4. Materials and Methods

### 4.1. Study Design and Participants

Patients were followed in the French Reference Center for Lysosomal Diseases. A total of 33 French patients with Gaucher Disease (GD) were recruited and assigned to 2 different groups. 15 GD patients under enzyme replacement therapy (ERT) were assigned to Group 1 (ERT GD) and 18 untreated GD patients were allocated to Group 2 (UT GD). Among the 18 participants from Group 2, 10 patients were enrolled in a 6 to 42 months longitudinal follow-up, after the first administration of ERT. None of the GD patients were splenectomized at the time of sampling. The patient’s characteristics and biological parameters are provided in Appendix A. 22.Healthy donors (CTL, Group 3) were recruited as controls. The study was conducted in accordance with the Declaration of Helsinki and was approved by the relevant local ethics committees (protocol 2014/55 NICB; *n*° IRB 00003835).

### 4.2. Blood Sampling

Blood samples were collected in EDTA containing tubes and were processed for freezing at −196 °C as described previously [50]. To perform erythrophagocytosis assay in a GD context, we collected RBCs from the blood sample. Peripheral blood mononuclear cells (PBMCs) were collected from GD patients and from healthy donors. The PBMCs were subjected to ficoll density gradient separation and were differentiated into macrophages following protocols provided in the Online Appendix A.

### 4.3. Erythrophagocytosis Assays

Thawed RBCs from healthy donors (CTL RBCs), untreated GD patients (UT GD RBCs), or ERT-treated GD patients (ERT GD RBCs) were labeled with carboxyfluorescein succinimidyl ester CFSE (ThermoFisher scientific) according to manufacturer’s instructions. Stained RBCs were incubated for 3 h at 37 °C at 5% CO_2_, at specific ratio effector/target cells (E/T) ratio, depending on the phagocytes. Non-phagocytized RBCs were then lysed and macrophages were analyzed by flow cytometry. CFSE^+^ macrophages were considered as cells having phagocytized at least one RBC. The PI was calculated and normalized using the ratio: geometrical mean fluorescence intensity (GeoMFI) of CFSE^+^ phagocytes divided by the GeoMFI of the CSFE-stained RBCs ×100. Internal controls were used to normalize data between different experiments. TO-PRO^®^-3 stain was used as a dead cell indicator. Heated RBCs (20 min at 50 °C) were used as a phagocytic positive control. Co-incubation performed at 4 °C, was regarded as a phagocytic negative control. Experiments were performed with THP1-derived macrophages at an effector/target cell (E/T) ratio of 1/100, or with blood derived-macrophages from healthy donors (E/T ratio 1/10). Blood-derived macrophages from healthy donors were treated or not with conduritol B epoxide (CBE; 1 mM) for 6 days (24 h after monocytes were processed) and the phagocytosis assay was done at an E/T ratio of 1/25. ROS production was assessed in similar erythrophagocytosis assays performed with undifferentiated THP1 cells (E/T ratio of 1/100). Macrophages’ phenotype analysis after erythrophagocytosis was performed following a 3 h, 12 h, or 24 h co-incubation time.

For the phagocytosis assays carried out using reticulocytes freshly generated and produced from an in vitro erythropoiesis in the presence or absence of CBE, we proceeded as described previously [8]. The erythroid differentiation was monitored from the pro-erythroblast stage to the reticulocyte stage for 18 days. On the 18th day, GPA **^+^** Hoescht^-^ cells were considered as reticulocytes and were sorted (MA900 Sony). Reticulocytes were labeled with CFSE and were incubated with THP1-derived macrophages (E/T ratio 1/10) for phagocytosis assay (3 h incubation). The PI was then calculated by multiplicating the % of CFSE^+^ macrophages by the CFSE Geomean.

### 4.4. Statistical Analysis

A parametric test (*t*-test) and non-parametric tests were used depending on whether or not the data followed a normal distribution. Mann-Whitney test or Wilcoxon signed-rank tests were used to compare unpaired and paired observations, respectively. Group comparisons between 3 conditions were performed using ANOVA and Kruskal-Wallis test. Correlations were assessed using Spearman or Pearson test. *p*-values < 0.05 were regarded as statistically significant (* *p* <0.05, ** *p* <0.01, *** *p* <0.001, **** *p* <0.0001).

## Figures and Tables

**Figure 1 ijms-23-07640-f001:**
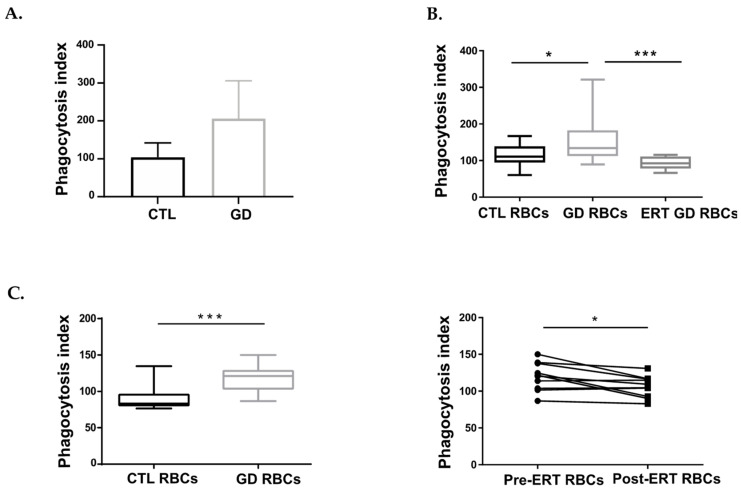
GD RBCs exhibit higher phagocytosis index (PI) by macrophages than CTL RBCs. (**A**) Erythrophagocytosis assays were performed using primary macrophages derived from healthy donor’s monocytes (CTL) or monocytes-derived macrophages from untreated GD patients (GD) and polarized toward the M2 phenotype. CTL macrophages were co-incubated with the autologous CTL RBCs (*n* = 3) and GD macrophages were co-incubated with the autologous GD RBCs (*n* = 3). The chart represents the PI of CTL and GD samples. The means and standard deviation are represented. (**B**) Erythrophagocytosis assays were performed using primary macrophages derived from healthy donor monocytes and polarized toward the M2 phenotype. The chart represents the PI of CTL samples (CTL RBCs; *n* = 15) and untreated GD samples (GD RBCs; *n* = 15), and treated GD samples (ERT GD RBCs, *n* = 8). Group comparison was performed using a Mann-Whitney test. The medians are represented as horizontal bars; the upper and lower quartiles are represented as the top and the bottom of the box, respectively; and the maximum and minimum data values are shown by dashes at the top and the bottom, respectively, of the whiskers. * *p* < 0.05; *** *p* < 0.001. (**C**) Left panel: Erythrophagocytosis assays were performed using THP1-derived macrophages co-incubated with CTL RBCs (*n* = 11) or GD RBCs (*n* = 18). The chart represents the PI of CTL and GD RBCs obtained for this cohort. Group comparison was performed using a Mann-Whitney test. The medians are represented as horizontal bars; the upper and lower quartiles are represented as the top and the bottom of the box, respectively; and the maximum and minimum data values are shown by dashes at the top and the bottom, respectively, of the whiskers. *** *p* < 0.001. Right panel: Erythrophagocytosis assays were performed using THP1-derived macrophages. The effect of ERT treatment on the PI was evaluated for RBCs from 10 patients before (Pre-ERT RBCs) and after (Post-ERT RBCs) ERT treatment (longitudinal study). Group comparison was performed using a paired *t*-test. * *p* < 0.05.

**Figure 2 ijms-23-07640-f002:**
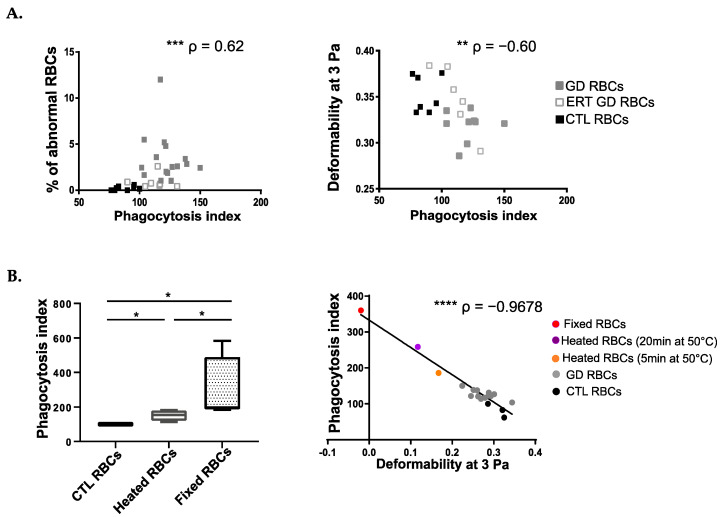
The uptake of GD RBCs correlates with cell deformability and morphology. (**A**) Left panel: Positive correlation between the PI and the percentage of abnormal morphologies of RBCs. Right panel: Negative correlation between the PI and the deformability of RBCs at 3 Pa. GD, ERT GD, and CTL RBCs values are depicted with gray, white and dark squares, respectively. The *p* and ρ values were determined using the Spearman rank correlation test. ** *p* < 0.01; *** *p* < 0.001. (**B**) Left panel: Erythrophagocytosis assays were performed using THP1-derived macrophages co-incubated with untreated CTL RBCs (*n* = 4) or CTL RBCs (*n* = 4) artificially made less deformable after heating for 20 min at 50 °C or after fixation with glutaraldehyde at 0.025% (*n* = 4). The chart represents the PI. The medians are represented as horizontal bars; the upper and lower quartiles are represented as the top and the bottom of the box, respectively; and the maximum and minimum data values are shown by dashes at the top and the bottom, respectively, of the whiskers. Group comparison 2 by 2 was performed using a Mann-Whitney test. * *p* < 0.05. Right panel: Negative correlation between the PI and the deformability of RBCs at 3 Pa (*n* = 19). CTL RBCs, GD RBCs, and CTL RBCs artificially made less deformable after heating for 5 min or for 20 min at 50 °C or after fixation with glutaraldehyde were used. The *p* and ρ values were determined using the Pearson correlation test. **** *p* < 0.0001.

**Figure 3 ijms-23-07640-f003:**
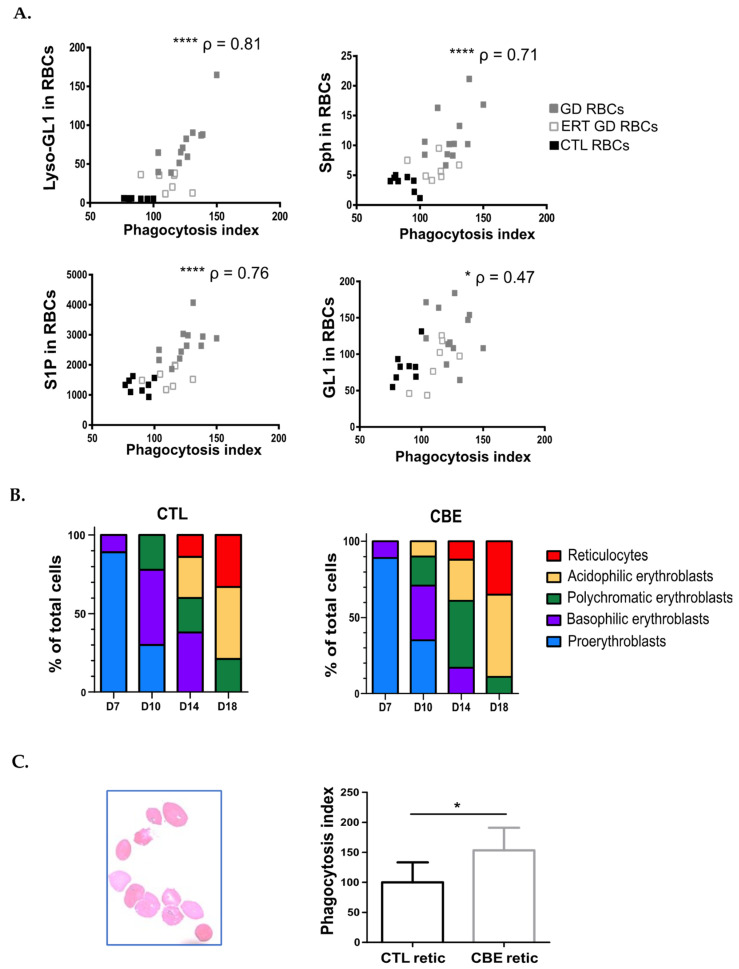
Sphingolipids overload of RBCs enhances their erythrophagocytosis. (**A**) Positive correlations between the PI and the sphingolipids content measured in RBCs (Lyso-GL1, Sph, S1P, and GL1). GD, ERT GD, and CTL RBCs values are depicted with gray, white and dark squares, respectively. The *p* and ρ values were determined using the Spearman rank correlation test. * *p* < 0.05; **** *p* < 0.0001. (**B**) Proportion of erythroid progenitors on days 7, 10, 14, and 18 of the erythroid differentiation performed without (left panel) or with CBE (right panel). Morphological analysis after May-Grünwald-Giemsa (MGG) staining was analyzed. (**C**) Left panel: Observation after MGG staining of reticulocytes sorted after 18-days of erythroid differentiation. Right panel: Erythrophagocytosis assays were performed using THP1-derived macrophages co-incubated with sorted reticulocytes obtained after 18 days of erythroid differentiation without (CTL retic; *n* = 5) or with (CBE retic; *n* = 5) CBE. The chart represents the PI. The means and standard deviation are represented. Group comparison was performed using a Mann-Whitney test. * *p* < 0.05.

**Figure 4 ijms-23-07640-f004:**
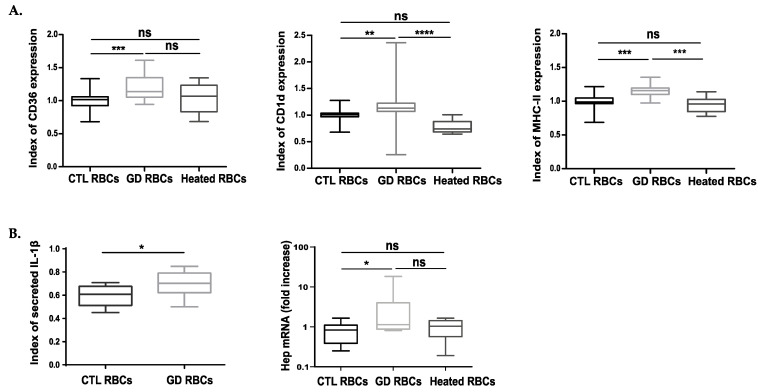
The phagocytosis of GD RBCs induces phenotypic modifications of macrophages toward Gaucher cells phenotype. (**A**) The expression of different markers was investigated at the surface of THP1-derived macrophages after 12 h of phagocytosis of CTL (*n* = 20), GD (*n* = 18), or CTL Heated (*n* = 8) RBCs. The expression levels of the phagocytic marker CD36, as well as the immunological markers MHC-II and CD1d antigen, were evaluated by flow cytometry. The charts represent the index of marker expression. Group comparison was performed using a Kruskal-Wallis test for CD1d and CD36 and an ordinary one-way ANOVA for MHC-II. ** *p* < 0.01; *** *p* < 0.001; **** *p* < 0.0001. ns = non-significant. (**B**) Left panel: Secretion levels of IL-1β were measured by ELISA. The charts represent the index of secreted cytokine in the supernatant of macrophages having engulfed CTL (*n* = 12) or GD (*n* = 13) RBCs. Group comparison was performed using an unpaired *t*-test. * *p* < 0.05. Right panel: hepcidin mRNA (Hep) quantification by qRT-PCR in macrophages 3 h after phagocytosis of CTL (*n* = 7) or GD (*n* = 7) RBCs. The charts represent the fold increase quantification of hepcidin mRNA compared to the quantification assessed in macrophages before erythrophagocytosis. Group comparison was performed using a Kruskal-Wallis test. * *p* < 0.05. ns = non-significant. The medians are represented as horizontal bars; the upper and lower quartiles are represented as the top and the bottom of the box, respectively; and the maximum and minimum data values are shown by dashes at the top and the bottom, respectively, of the whiskers.

**Figure 5 ijms-23-07640-f005:**
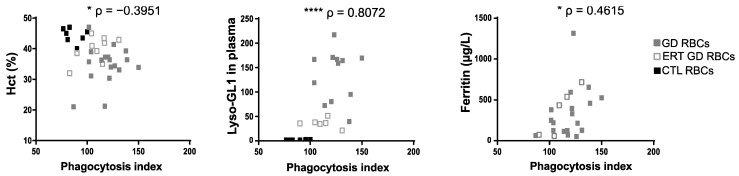
The enhanced phagocytosis of GD RBCs correlates with markers of disease activity. The PI of RBCs negatively correlated with the percentage of the hematocrit (Hct) (left panel) and positively correlated with the Lyso-GL1 (middle panel) and ferritin (right panel) plasmatic concentration. GD, ERT GD, and CTL RBCs values are depicted with gray, white and dark squares, respectively. The *p* and ρ values were determined using the Spearman rank correlation test. * *p* < 0.05; **** *p* < 0.0001.

## Data Availability

The authors did not report any data.

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
