# Peer review of "Phagocytosis of Erythrocytes from Gaucher Patients Induces Phenotypic Modifications in Macrophages, Driving Them toward Gaucher Cells"

_ijms, 2022, doi:10.3390/ijms23147640_

Round 1

Reviewer 1 Report

Thank you for the opportunity to review the manuscript. The manuscript is generally well written.

Few comments-

Introduction- please clarify what you meant in the sentence- "GC as well as macrophages from Gaucher’s patients"- GC are macrophages also. No?

Results:

2.1- What were the Hct/Hemoglobin levels of the samples used from patients with GD? Please explain if and how the Hct/Hemoglobin levels can affect the results.

Figure 1A- p-value is missing.

Figure 2A- suggest changing the scale of y-axis to better show the correlation. It seems that the more significant correlation is for the GD-RBC is this true?

Figure 5- very interesting results-can the correlation with lyso-GL1 adjusted for hematocrit and ferritin levels?  

Reviewer 2 Report

Congratulations to the authors for the findings set out in this article; they are very interesting and to continue to be explored for the scientific community.

To date, the actors of Gaucher disease are considered lipid-laden macrophages that turn into "Gaucher cells" with a striated appearance and off-center nuclei and that are considered the mechanism underlying the phenotypic alterations found in patients with GD. Nevertheless I agree that GD RBCs could play a more important role in pathogenesis than previously thought by interfering with key physiological processes leading to hallmark features in GD. Furthermore erythrophagocytosis of lipid-laden GD RBCs inducing phenotypicalterations in normal macrophages, up-regulating the expression of inflammatory molecules and hepcidin should also explain the increase in ferritin in the blood found in many patients with Gaucher disease where it is hypothesized that one of the causes of hyperferritinemia in these patients is the high serum levels of hepcidin.

The authors in the discussions could better deepen this link between erythrophagocytosis hepcidin and hyperferritinemia.

Minor modifications:  try to avoid abbreviations in the abstract

Reviewer 3 Report

Minor points:

Across the whole manuscript macrophages are described and treated as one entity. It is well known that there are several subsets. This topic has been widely described, even in the context of lysosomal storage diseases (PMID: 33919858). This should be mentioned in the introduction and discussion. In this sense, what subtypes of macrophages are used in this project?

Lipid laden macrophages are a hallmark of many LSDs, including NPC, NPA, Fabry, and atherosclerosis (PMID: 34660203, PMID: 26707209, PMID: 23452955, PMID: 17084825). Increased phagocytosis has been shown in NPC microglia, a macrophage-related cell type (PMID: 33627648). The results shown in this manuscript should be discussed along with their implications to other LSDs and atherosclerosis.

Since many years gadolinium salts have been used to inhibit the endocytosis of macrophages (reviewed in PMID: 11314600). I believe that this could be an interesting control for this study because it can propose a novel therapeutic strategy for GD which was tested in an NPC mouse model (PMID: 30441844).

An important advantage of using CBE is that it can be washed-out, and changes (reversibility) of the phenotypes can be assessed in the same cells (paired study). This is different from what was done with the ERT experiments because they were performed in vivo, in patients, therefore they are different cells. This can expend the projections of their studies and I believe it should be done.
